

# The influence of mesoscale climate drivers on hypoxia in a fjord-like deep coastal inlet and its potential implications regarding climate change and greenhouse gas production: examining a decade of water quality data

Johnathan Daniel Maxey[12], Neil D. Hartstein[2], Aazani Mujahid[3], Moritz Müller[1]

[1]Faculty of Engineering, Computing and Science, Swinburne University of Technology, Kuching 93350, Malaysia
[2]ADS Environmental Services, Kota Kinabalu, Sabah, 88400, Malaysia
[3]Faculty of Resource Science & Technology, University Malaysia Sarawak, Kota Samarahan 94300, Sarawak, Malaysia

*Correspondence to*: Johnathan Daniel Maxey (jdaniel@adseser.com)

**Abstract.**

Deep coastal inlets are sites of high sedimentation and organic carbon deposition that account for 11% of the world's organic carbon burial. Australasia's mid to high latitude regions have many such systems. It is important to understand the role of climate forcings in influencing hypoxia and organic matter cycling in these systems, but many such systems, especially in Australasia, remain poorly described.

We analysed a decade of *in-situ* water quality data from Macquarie Harbour, Tasmania, a deep coastal inlet with more than 180,000 tons of organic carbon loading per annum. Monthly dissolved oxygen, total Kjeldhal nitrogen, dissolved organic carbon, and dissolved inorganic nitrogen concentrations were significantly affected by rainfall patterns. Increased rainfall was correlated to higher organic carbon and nitrogen loading, lower oxygen concentrations in deep basins, and greater
oxygen concentrations in surface waters. Most notably, the Southern Annular Mode (SAM) significantly influenced oxygen distribution in the system. High river flow (associated with low SAM index values) impedes deep water renewal as the primary mechanism driving basin water hypoxia. Climate forecasting predicted increased winter rainfall and decreased summer rainfall, which may further exacerbate hypoxia in this system.

Currently, the Harbour basins experience frequent (up to 36% of the time) and prolonged (up to 2 years) oxygen-poor conditions with the potential to promote greenhouse gas ($CH_4$, $N_2O$) production. Increased greenhouse gas production will alter the processing of organic matter entering the system. The increased winter rainfall predicted for the area will potentially increase greenhouse gas emissions due to increased spread and duration of hypoxia in the basins. Further understanding of these systems and how they respond to climate change will improve our estimates of future organic matter cycling (burial vs
export) and greenhouse gas production.



# 1 Introduction

Fjords and fjord-like estuaries (also called Deep Coastal Inlets – DCI; Keith *et al.* 2020) are sites of high sedimentation and organic carbon (OC) burial. These systems account for approximately 11% of the world's annual OC burial (Smith *et al.* 2015). Compared to other marine benthic environments (*e.g.* sediments along the continental shelf, deeper pelagic sediments,

shallow-water carbonate sediments), they bury the most OC per unit area (Smith *et al.* 2015; Bianchi *et al*. 2018, 2020).

Their location within mid to high latitude coastal margins and disproportionate role in geochemical cycling make these systems especially vulnerable to anthropogenic pressure (Walinsky *et al.* 2009; Gilbert *et al.* 2010; Bianchi *et al.* 2018, 2020). Bianchi *et al.* (2018) have classified fjord and fjord-like DCIs as "Aquatic Critical Zones" in need of further

investigation, especially regarding how they might respond to changes in climatological drivers and anthropogenic pressure. One of the critical issues facing coastal environments is the expansion of poor oxygen conditions due to increased anthropogenic organic matter loadings (Diaz and Rosenberg 2008; Oschlies *et al.* 2018; Breitburg *et al*. 2018; Pitcher *et al.* 2021).

Combined effects of environmental drivers or forcings drive the distribution of dissolved oxygen (DO) in any given system. In fjord and fjord-like DCIs, this includes wind, tidal exchange, river flow, organic loading, deep water renewal (DWR) (Edwards and Edelsten 1977; Gade and Edwards 1980; Geyer and Cannon 1982) and microbial processing in the sediments and water column (Gilibrand *et al*. 2006; Maxey *et al.* 2020). Characteristics of these systems are shallow sills at their mouth and several sills or ridges that separate the estuary into various basins (Pickard and Stanton 1980; Stanton and Pickard 1980

Inall and Gillibrand 2010). These morphological features restrict mixing and promote stratification of the water column by isolating basin waters from exchange mechanisms between the coastal ocean and surrounding catchment (Inall and Gillibrand 2010). Hypoxia (*defined as* DO concentrations below 2 mg L$^{-1}$) has been long recognised and can be a natural feature of these systems (Rosenberg 1977; Rabalais *et al*. 2010, Inall and Gillibrand 2010; Ji *et al*. 2020).

The availability of DO influences the eventual fate of OC processed by microbial communities as it enters either aerobic or anaerobic metabolic pathways (*see* del Giorgio and Williams 2005). In addition, the cycling of organic matter and nutrients in poor oxygen environments often leads to the production of potent greenhouse gasses such as methane ($CH_4$) and nitrous oxide ($N_2O$) (Codispoti *et al.* 2005). The fate of carbon exported to marine systems from estuaries is tied to oxygen distribution. Estuarine morphology, physical oceanography, and anthropogenic impacts (*e.g.* hydroelectric dams, land-use

modification, sewage outfalls, etc.) drive the oxygen distribution.

Understanding how DO distribution and availability in fjord-like systems respond to climate change requires understanding how it currently responds to changes in local and mesoscale environmental drivers. These drivers include rainfall and runoff





and associated nutrient and organic matter loading. Predicted climate change impacts include changes in air pressure, wind
strength, rainfall patterns, and storm intensity (Grose *et al.* 2010; Priestley and Catto 2021; Goyal *et al.* 2021), all of which
have the potential to affect DO distribution in fjord-like estuaries (Gillibrand *et al.* 2005, 2006; Austin and Inall 2011;
Hartstein *et al.* 2019).

Understanding how broader environmental drivers affect localised DO distribution requires spatially extensive long-term
datasets which are not readily available in many systems. Ideally, these datasets would provide enough statistical power to
tease out relationships between external drivers (*e.g.* rainfall volume, rainfall accumulation, OC and organic nitrogen (ON)
loading, river flow and climate oscillation indices) and DO distribution through the water column. A 10-year dataset is
available for a relatively remote DCI on Tasmania's West Coast, enabling analyses focusing on long-term trends in water
quality, freshwater and organic matter loading, and their relationships with climate drivers.

The aims of this paper are to:

1. Understand the effects of rainfall or freshwater inputs on OM loading, nutrient loading, and DO distribution in a
fjord-like deep coastal inlet.

2. Describe the current effects that broader climate oscillations have on DO distribution and discuss implications
future climate predictions have on possible DO dynamics in these systems (example of a restricted sill system),
especially regarding physical drivers of deep water renewal.

3. Discuss implications for managing these systems regarding the regulation of freshwater input, OM loading, and
the potential for GHG emissions.

## 2 Methods

### 2.1 Study Area

Macquarie Harbour is a fjord-like DCI located on the West Coast of Tasmania. Although it's glacially carved status of the
Harbour itself is somewhat unclear (Baker and Ahmad 1959; Kiernan 1990, 1991,1995), it has the morphology and resulting
oceanographic dynamics shared by many fjords and fjord-like systems, including a propensity for oxygen-poor basins
(Creswell *et al.* 1989; Hartstein *et al.* 2019). Descriptions of its DO drivers (Hartstein *et al.* 2019 and Maxey *et al.* 2017,
2020) suggest disparate processes affecting surface water and basin DO distribution. Namely, DO in the basin waters is



resupplied by DWR. Where or when the direct effects of these processes wane, diffusive mixing and water column oxygen demand become the key drivers of oxygen availability.

The Harbour is oriented in an NW by SE direction, is approximately 33 km long, 9 km wide, and has a surface area of 276 km². Compared to the rest of Australia, Western Tasmania receives some of the highest rainfall (more than 2,500 mm year⁻¹)
and high seasonal rainfall variability (Dey *et al.* 2018). As a result, broad-scale climate oscillations like the Southern Annual Mode (SAM) (Meneghini, Simmonds, and Smith 2007; Hill, Santoso, & England, 2009) affect westerly winds that generate orographic rainfall in Macquarie Harbour's catchment. Since the 1970s, both the SAM index (positive values associated with stronger westerlies) and winter rainfall in Macquarie Harbour's catchment have increased (Taschetto & England 2009; Marshall *et al.* 2018; Fogt and Marshall 2020).


The primary source of fresh water to the Harbour is the Gordon River which is responsible for up to 82% of the system's freshwater input (Hartstein *et al.* 2019). The mouth of the Gordon River is located on the Harbour's SE end and drains a combined catchment (including the Franklin River) of 5,682 km² (MHDOWG 2014). This catchment is located west of the Cradle Mountain Range (**Figure 1**). At the NW end of the Harbour, the King River is the second largest contributor of fresh
water to the system and drains a catchment area of 802 km².

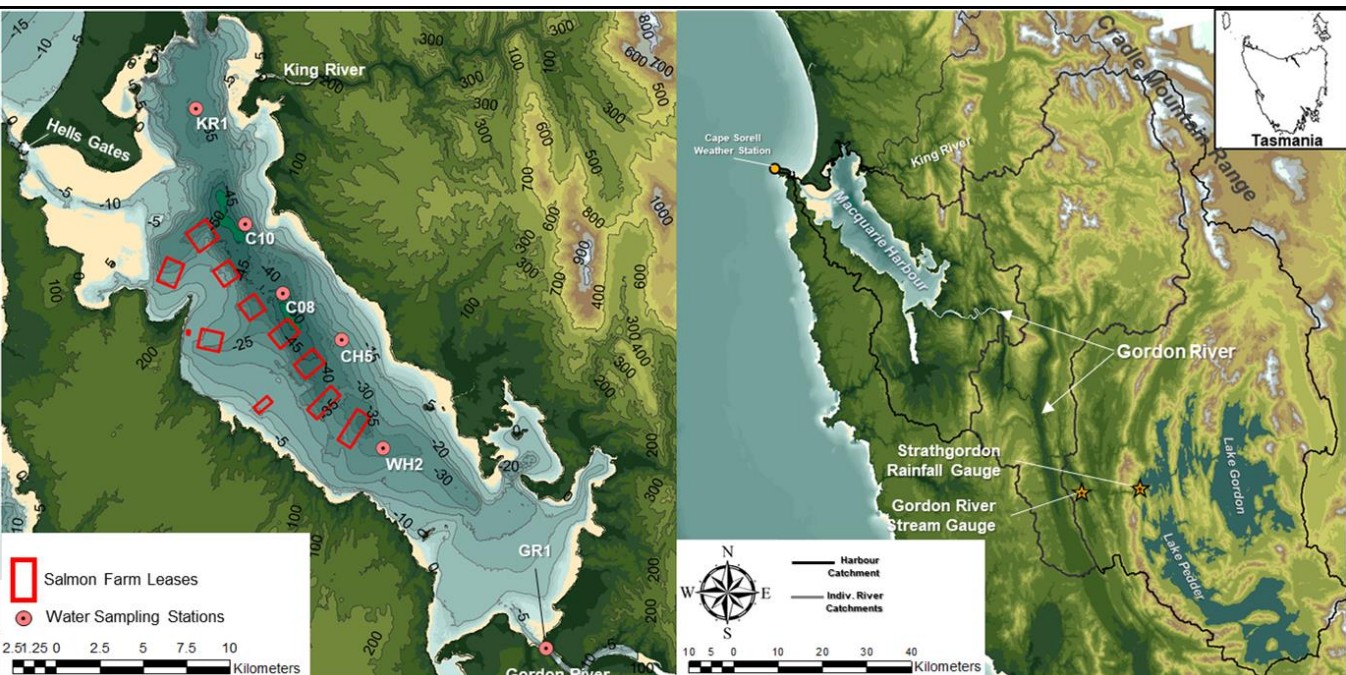

**Figure 1: Macquarie Harbour, Tasmania. Monthly sampling sites shown with red circles, Gordon River Stream Gauge and Strathgordon Rainfall Gauge shown as orange stars. Aquaculture lease boundaries are shown as hollow red rectangles. Lease locations are sourced from Land Information Systems Tasmania (LISTmap - https://maps.thelist.tas.gov.au/). Station names**



**reflect general harbour locations where KR1 indicates King River 1; C10, C08, CH5 indicate Central Harbour 10, 08, 5; WH2 indicates World Heritage Area 2; and GR1 indicates Gordon River station 1.**

River flow in the upper catchments of the Gordon and King rivers has been regulated by hydroelectric dams since 1983 (operated by Hydro Tasmania). Gordon River mean flow has been reported to range from 190 to 265 $m^3$ $sec^{-1}$ and the King

River at approximately 55 $m^3$ $sec^{-1}$ (Carpenter *et al*. 1991; Koehnken 2006). More recent analyses indicate that mean monthly flows can vary significantly, with flows < 100 $m^3$ $sec^{-1}$ observed in summer and early autumn and 500 $m^3$ $sec^{-1}$ in late autumn, winter, and spring (Hartstein *et al*. 2016; Maxey *et al*. 2020). Daily peak flow can reach over 1500 $m^3$ $sec^{-1}$. However, before the introduction of hydroelectric dams, peak flow could have been greater (King 1980; King and Tyler 1981; King and Tyler 1982; Walker 1985).


Both rivers supply dark, tannin-rich waters, limiting the system's photic zone to the first 2-3 m of the water column (Carpenter *et al*. 1991; Edgar *et al*. 1999). These tannin-rich waters are a product of high organic loading, estimated at up to 180,000 tons OC per year (Hartstein *et al*. 2016; Maxey *et al*. 2020). While the Harbour's catchments are largely undeveloped native forest, anthropogenic inputs include treated copper mining discharges from the King River and several

commercial salmon farms situated within the Harbour's centre. Salmon farming has been present in the Harbour since the 1980s. Between 2009 and 2016, it underwent significant expansion from 9000 tonnes to 16,000 tonnes. Since 2016 farm biomass has been set at 9,500 tonnes (Dept Natural Resources and Environment Tasmania https://salmonfarming.nre.tas.gov.au) (see **Figure 1** for an overview of farming leases).

Macquarie Harbour has a strongly stratified water column set up by a long (14 km), shallow (< 3m in areas) constricted sill and oxygen-poor basins (Cresswell *et al*. 1989; Maxey *et al*. 2017). Temperature differences above and below the halocline can be 10 °C. However, temperatures in the bottom waters exhibit little variability (2 to 3 degrees) (Hartstein *et al*. 2019). Basin waters are refreshed episodically by DWR, which is driven by a combination of low atmospheric pressure, low harbour water level, and sustained NW winds (Hartstein *et al*. 2019). However, the relative importance of freshwater supply

to DWR and oxygen distribution remains unknown.

Climate change models reviewed and presented in Grose *et al*. (2010) and Bennett *et al*. (2010) indicate that the west coast of Tasmania, including the main rainwater catchments feeding Macquarie Harbour, is expected to experience increased storm intensity, wetter winters, and drier summers (up to 10% to 18% wetter or drier throughout the catchment). However,

there has been a steady increase in spring and winter rainfall in Western Tasmania (Taschetto & England 2009) despite increasing SAM index values (Marshall *et al*. 2018). There is still a need to clarify the specific impacts of a changing climate on rainfall patterns. Nevertheless, to predict how future climate scenarios might affect harbour dynamics, it is vital to understand how the system currently responds to present drivers.





Much of the previous literature describing the oceanography and environmental status of Macquarie Harbour has focused on the effects of copper mine discharge (Carpenter *et al*. 1991; Koehnken 1996; Featherstone *et al*.1997; Stauber *et al.* 2000; Eriksen *et al*. 2001; Teasdale *et al*. 2003, Augustinus *et al*. 2010; Cracknell *et al*. 2019. Recently, Hartstein *et al*. 2019 and Maxey *et al*. 2020) have expanded the understanding of the physical and biological oceanography (initially described by Cresswell *et al*. 1989) of the Harbour. They found that DWR is the major driver of bottom water oxygen distribution and that

Gordon River organic loading is the primary driver of pelagic oxygen demand (POD). Da Silva *et al*. (2021) examined the microbial communities present in the Harbour's water column. They showed distinct functional groups along salinity and depth gradients, suggesting that external climatic drivers influence harbour processes.

**2.2 Data Collection and Analysis**

Water quality data are available from a monthly water quality monitoring program (since October 2011). The monitoring data examined here included sonde profiles and *in-situ* water samples taken at several sites within the Harbour (**Figure 1** and **Table 1**). These include stations at end members and within each of the Harbour's basins. After its initial stages (2011 to 2013) the sampling program was expanded to include additional sites and parameters (**Table 1**).

Water quality sonde profiles were collected every meter using a YSI-6600 V2 equipped with optical DO, salinity, temperature, and depth sensors. Sonde calibration was checked and corrected (when needed) each sampling period. Water samples were collected at various depths (see **Table 1**) using a 5 L Niskin bottle sampler. Water sample parameters include total organic carbon (TOC) and dissolved organic carbon (DOC), total Kjeldahl nitrogen (TKN), soluble ammonia ($NH_3$), nitrate ($NO_3$), and chlorophyll-a. Water collected for soluble inorganic N was filtered immediately using 0.45 µm

polyethersulfone syringe filters (Whatman Puradisc), and all samples were stored in a chilled dark container until being transported to the lab for analysis.

Analytical Services Tasmania analysed all water samples. Maxey *et al*. (2020) outlined detailed organic carbon and chlorophyll-a methodologies. Dissolved $NH_3$, $NO_3$, and TKN were analysed using a Lachat Flow Injection Analyser. $NH_3$

and $NO_3$ analyses used methods based on APHA Standard methods (2005) 4500-$NH_3$ H (reporting limit 0.005 mg $L^{-1}$) and 4500 - $NO_3$ I (reporting limit 0.002 mg $L^{-1}$). TKN was determined by converting N into $(NH_4)_2SO_4$ using $K_2SO_4$ digestion and reacting this digested sample with alkaline buffer, salicylate and hypochlorite to form a coloured compound to be read on the Lachat analyser (reporting limit 0.1 mg $L^{-1}$).




**Table 1: Monthly sampling stations showing coordinates, starting months of individual parameters, and sampling depth (in meters).**

| Station | Dissolved Oxygen / Salinity | NPOC | TKN | NH$_3$ | NO$_3$ | CHL- a |
|---|---|---|---|---|---|---|
| **KR1** 361316, 5325972 | Oct 2011 *Every Meter* | - | July 2014 2, 10, 20, B2 | Oct 2011 2, 10, 20, B2 | Oct 2011 2, 10, 20, B2 | Dec 2011 2, 12 |
| **C10** 363708, 5320464 | Dec 2013 *Every Meter* | July 2014 *Every Meter* | Dec 2013 2, 10, 20, B2 | Dec 2013 2, 10, 20, B2 | Dec 2013 2, 10, 20, B2 | Sep 2014 2 |
| **C08** 365489, 5317238 | Dec 2013 *Every Meter* | - | Sep 2014 20, B2 | Dec 2013 20, B2 | Dec 2013 20, B2 | Dec 2013 2, 12 |
| **CH5** 368215, 5315124 | Oct 2013 *Every Meter* | - | Oct 2013 1, 2, 10, 20, B2 | Oct 2013 1, 2, 10, 20, B2 | Oct 2013 1, 2, 10, 20, B2 | Oct 2013 2, 12 |
| **WH2** 370218, 5309894 | Oct 2011 *Every Meter* | July 2014 1, 2, 5, 10, 15, 20, 30, B2 | Oct 2011 1, 2, 5, 10, 15, 20, 30, B2 | Oct 2011 1, 2, 5, 10, 15, 20, 30, B2 | Oct 2011 1, 2, 5, 10, 15, 20, 30, B2 | Dec 2011 2, 12 |
| **GR1** 377784, 5300603 | July 2014 *Every Meter* | July 2014 2, B2 | Dec 2013 2, B2 | Dec 2013 2, B2 | Dec 2013 2, B2 | Dec 2013 2 |

*Station coordinates given in UTM (Zone 55G)*
*Dates indicate starting month of sampling parameter*
*Sampling depth indicated under each date with B2 = 2m from seabed*

The Bureau of Meteorology (BOM) provided rainfall and stream gauge data from several gauging locations within the Gordon River catchment, namely Strathgordon rainfall gauge station and the Gordon Above Denison stream gauge (hereafter referred to as "Gordon River Stream Gauge"; **Figure 1**). Daily rainfall and streamflow data were available for the entire span of the monthly water quality program, and the Strathgorden rainfall dataset extends back until the 1970s. Rainfall data were organised into several metrics, including daily average monthly rainfall and total accumulated rainfall 30, 20, 10, 5, 3, 2 days

and one day before each monthly monitoring sampling period. The seasonality of rainfall was analysed using a two-way ANOVA grouping data by year and season. *Post hoc* analyses were performed using TUKEY HSD.

The flow was estimated at the mouth of the Gordon River (for loading estimations) by scaling daily rainfall to the size of the catchment and assuming a rainfall and runoff coefficient of 0.70 adopted from a neighbouring catchment with similar land

cover, geology, and slope (Willis 2008). Additional streamflow from Gordon River dam releases was estimated by





subtracting scaled rainfall contributions to river flow measured at the Gordon River Stream Gauge. This flow was added to the estimated runoff entering the Harbour. OC and N loading from the Gordon River was determined by multiplying the parameter concentration by the estimated flow entering the Harbour.

The relationship between rainfall (of various metrics), riverine OC and N loading, and water quality (*e.g.* DO concentration and salinity) was analysed using Pearson correlations for each site, for each 1 m depth bin, and the entirety of the water quality dataset. In heavily stratified systems, the surface layer's depth depends upon freshwater supply, tidal forcing, and atmospheric pressure (*i.e.* thicker freshwater lens during flood events) (Gillibrand *et al.* 2005; Cage *et al.* 2006; Inall and Gillibrand 2011). When comparing sonde profiles over multiple sampling periods, surface referenced data have a shifting

datum due to the varying thickness of the freshwater lens and thus introduce a source of error in the analysis. To reduce this possible source of error, we arranged the water quality data to reference height from the seabed (not depth from the surface) when comparing it to catchment rainfall and river flow.

Hypoxic volume was estimated by scaling up monthly DO concentrations using a 1 m vertical resolution box model. Oxygen

concentrations within each box were represented by data from individual monitoring stations located in the boxes. A representative volume was assigned to each station's box by using the harbour hypsography in 1 m bins (bathymetry data were provided by Lucieer *et al.* 2007 and Hartstein *et al.* 2016). To estimate the hypoxic volume in the system, we assumed DO concentrations within the box did not vary. DO concentrations below 10 m depth were then multiplied by each bin's representative volume of water.

**3 Results**

**3.1 Rainfall Patterns (2011-2021)**

Significant seasonal patterns in rainfall were observed at the Strathgordon station with greater rainfall (daily average and accumulated) in winter (June - August) (daily average $12.9 \pm 0.9$ mm day$^{-1}$) relative to summer (December - February) ($6.6 \pm 0.6$ mm day$^{-1}$) and autumn (March - May) ($8.3 \pm 0.7$ mm day$^{-1}$) (see **Figure 2** and **Table 1**). *Post hoc* testing could not

resolve a significant difference between spring (September - November) ($10.6 \pm 0.9$ mm day$^{-1}$) and the other seasons. Total accumulated rainfall before sampling also showed significant seasonal patterns. There were no significant interannual effects on rainfall of any metric.





Figure 2: Box plots of flow and rainfall metrics from 2011 to 2021 including (A) Gordon River Stream Gauge flow, (B) Estimated daily flow into the Harbour, (C) Daily average rainfall at Strathgordon station, (D) 30 day, (E) 20 day, (F) 15 day, (G) 10 day, and (H) 5-day rainfall accumulation before monthly water quality sampling. *Post hoc* (Tukey HSD) grouping is shown above the plot area for datasets where two-way ANOVA revealed a significant seasonal effect.

The average daily river flow measured from the Gordon River Stream Gauge (see **Figure 1**) ranged from 17 m$^3$ sec$^{-1}$ to 263 m$^3$ sec$^{-1}$ (**Figure 2**). Two-way ANOVAs revealed a significant seasonal and yearly effect (with significant interaction). Minimum daily average flows ranged from 17 m$^3$ sec$^{-1}$ in 2012 to 117 m$^3$ sec$^{-1}$ in 2013. Hydroelectric releases, and not catchment rainfall, would predominantly determine flow at the Gordon River Stream Gauge as the upstream catchment area





between the dam and the gauge is relatively small (only 49.5 km$^2$, or about 0.8% of the size of the ungauged catchment area

feeding the Gordon River).

Estimated river flow into the harbour also exhibited statistically significant ($p = 0.04$) seasonal periods and significant ($p = 0.01$) interannual variation. The average winter period daily flow was estimated to be 759 ($\pm$ 67) m$^3$ sec$^{-1}$. During summer, the estimated flow was approx. 502 ($\pm$ 62) m$^3$ sec$^{-1}$. The greatest estimated flow observed occurred in August 2018 at approx.

1,630 m$^3$ sec$^{-1}$.

### 3.2 Rainfall Patterns and Water Quality at the Gordon River Mouth

At the mouth of the Gordon River, DOC and DON concentrations ranged from 4 mg L$^{-1}$ to 21 mg L$^{-1}$ in surface waters and 2 mg L$^{-1}$ to 19.8 mg L$^{-1}$ 2 m from the riverbed. Correlations between daily average rainfall and OC concentrations were

statistically significant but the effect size was stronger with samples taken 2 m off the seabed (r = 0.70, $p$ = 1.31 x 10$^{-13}$) compared to surface water correlations (r = 0.53, $p$ = 1.53 x 10$^{-7}$; **Figure 3**). The relationship between OC and rainfall accumulation was strongest five days before sampling (r = 0.7 OC). It should be noted that the correlation coefficient between rainfall and bottom water OC concentration was relatively stable (and above 0.6) from 5 to 30 days of accumulation before sampling.


DON concentrations at station GR1 ranged from 0.042 mg L$^{-1}$ to 0.43 mg L$^{-1}$ in surface waters and from 0.134 mg L$^{-1}$ to 0.46 mg L$^{-1}$ 2 m from the riverbed. As observed with the OC samples, ON concentrations were significantly correlated with rainfall, although the strength of this relationship was more modest in comparison. Daily average rainfall correlation coefficients from surface and bottom samples ranged from r = 0.33 ($p$ = 0.0062) to 0.47 ($p$ = 8.0 x 10$^{-5}$) respectively. The

strongest correlation observed between rainfall and DON concentration was found using rainfall accumulation five days prior to sampling (r = 0.65).

Rainfall and estimated flow were significantly correlated with TKN (greatest observation was r = 0.65, 5 days accumulation) and NO$_3^-$ concentrations (greatest was r = -0.64 10 days prior to sampling). The nature of the relationships between rainfall

and NO$_3^-$ concentration differed based on water sampling depth. Correlations between rainfall and NO$_3^-$ concentrations 2 m off the riverbed were positive (strongest relationship r = 0.49, 5 days accumulation before sampling), but surface water concentrations were negatively correlated (strongest relationship r = -0.64, 10 days accumulation before sampling). No significant relationships were observed between rainfall / estimated river flow and NH$_3$.





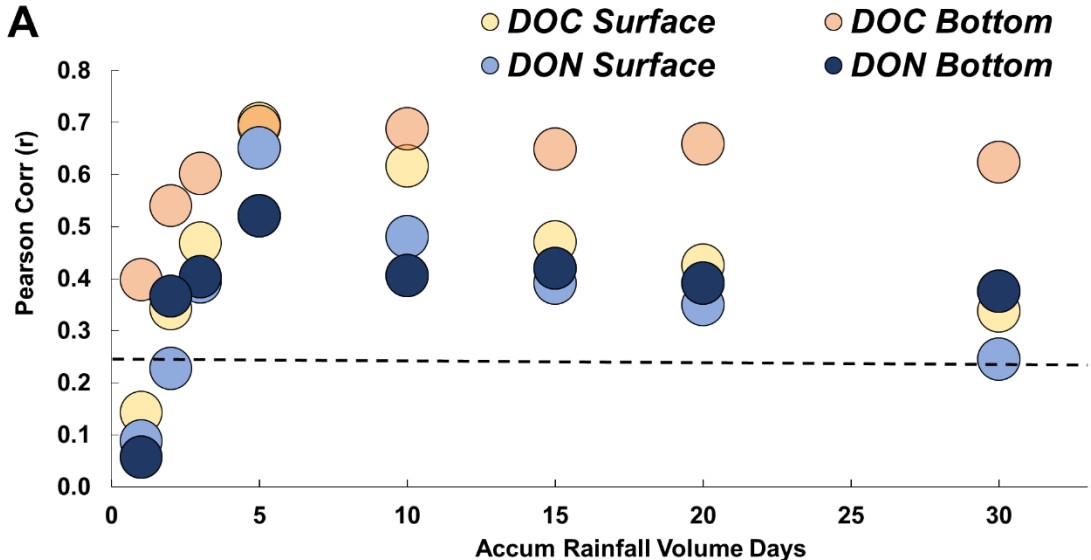

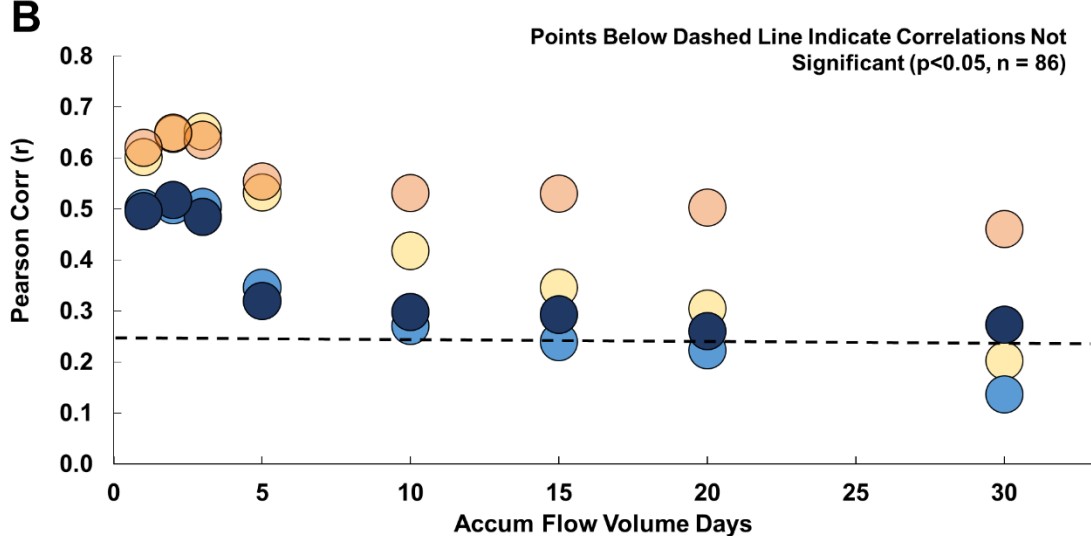

**Figure 3: Pearson Correlation Coefficients (r) (n = 86) between dissolved organic C and dissolved organic N concentrations measured at the Gordon River mouth and (A) accumulated rainfall (B) accumulated flow at the Gordon River mouth. Note that all correlations are statistically significant (p < 0.05) except those circles whose centres are below the dashed line.**

DOC loading (calculated from samples collected at 2 m depths and 3-day flow accumulation) was observed to significantly (p = 0.0478) vary with season. Winter (June to August) loadings (mean loading 642.8 ± 48.0 tons day$^{-1}$) were substantially greater than summer (December to February) loadings (mean loading 430.4 ± 57.8 tons day$^{-1}$) (**Figure 4**). The maximum OC loading rate was observed in May 2016 at 1,227 tons OC day$^{-1}$.





**Figure 4: Daily OC and ON loading (tons) for each month based on estimated daily flows and concentrations of OC and ON measured at the Gordon River mouth from 2015 to 2021. Note the large variation in the month of May is driven by large flows in years 2015 and 2016. Dashed line indicates average daily OC and ON loading from farm feed inputs based on 10,000 tonnes annual production, feed conversion rate = 1.3, assimilation rate of 85%, 49% feed carbon content, 6.7% feed nitrogen content.**

**3.3 Dissolved Oxygen Distribution**

Dissolved oxygen concentrations in Macquarie Harbour exhibited strong stratification and ranged from suboxic ($< 1\text{mg L}^{-1}$) in the deep basins to over 10 mg $L^{-1}$ at the surface (**Figure 5**). Below the surface lens of well-oxygenated water, there was a





mixing zone of approximately 10 m where DO concentrations rapidly declined, and salinity increased. Consequently, the waters below the mixing zone (referred to as basin water) were more saline and DO poor than the surface waters.


Harbour basin waters closest to the ocean endmember generally had higher DO concentrations (*i.e.* means of approximately 4 mg L$^{-1}$ observed at stations KR1 and C10; **Figures 5** and **6**). Towards the Gordon River, basin water DO concentrations were lower and were more often observed to be hypoxic (< 2mg L$^{-1}$) or suboxic (< 1mg L$^{-1}$). At stations near the ocean

endmember (also the Harbour's deepest basins; C10 and C08), the DO concentration and salinity were often observed to increase with depth. This water mass is the product of DWR (see Hartstein *et al.* 2019).

Our box model of Harbour basin waters indicates there is often hypoxic (<2 mg L$^{-1}$) water present in the system and prolonged periods of suboxia (<1 mg L$^{-1}$) (**Figure 7**). The most prolonged period of sustained suboxia occurred from June

2015 to May 2017. The largest volume of suboxic water was observed in May 2014 at over 60 x 10$^6$ m$^3$. The frequency of hypoxic and suboxic observations at any one station increased with proximity to the Gordon River mouth. For example, while station C10 experienced suboxia 13.8% of the time (of 96 months), station WH2 experienced suboxic conditions in nearly 37% of observations (of 120 months; **Figure 6**).



**Figure 5: Box plots of dissolved oxygen concentration (mg L⁻¹) and salinity monthly profiles. Extent of dataset for each site is specified above the plots. Note vertical resolution is 1 m and references height from the seabed. Hypoxic (< 2 mg L⁻¹) conditions are highlighted in red.**



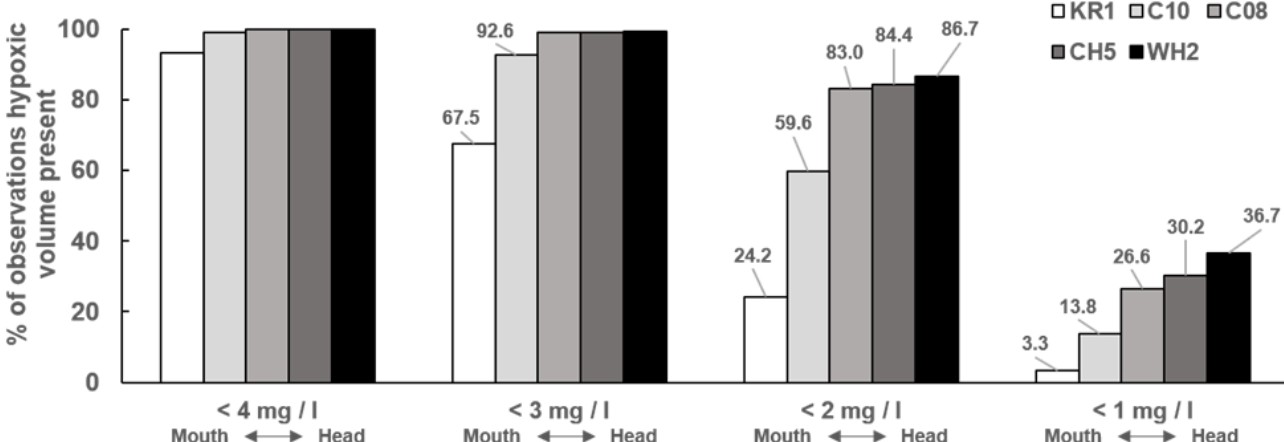

**Figure 6: Percentage of total observations where monthly sonde profiles contained dissolved oxygen measurements less than 4 mg L$^{-1}$, 3 mg L$^{-1}$, hypoxic water (<2 mg L$^{-1}$), and suboxic (<1 mg L$^{-1}$) water at any depth.**

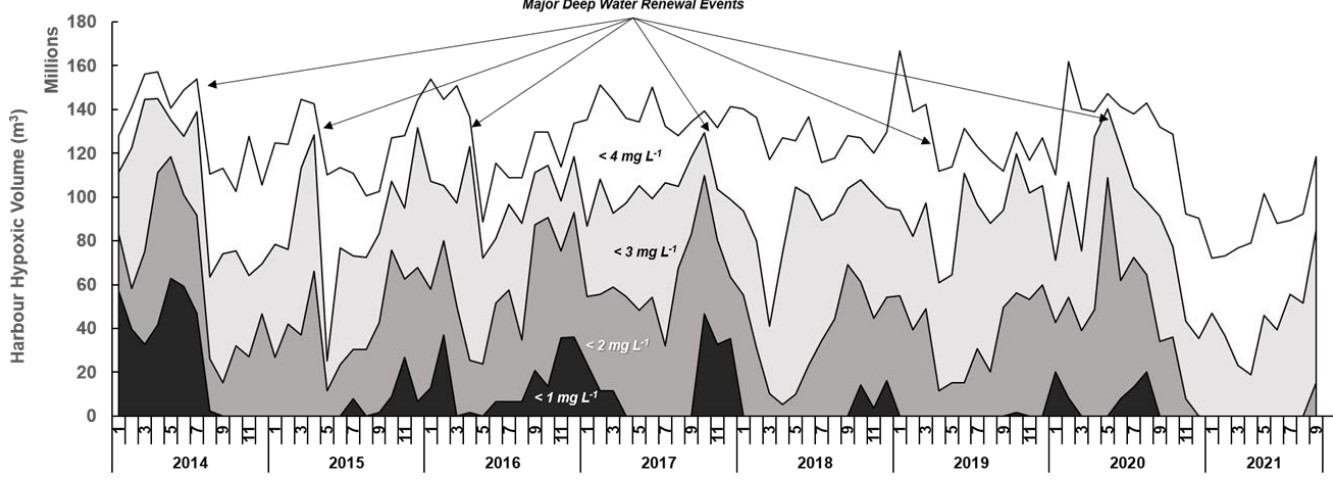

**Figure 7: Estimated hypoxic volume present in the Harbour from 2014 to 2021, based on our 1 m resolution box model. Deep water renewal events are denoted by black arrows.**


## 3.4 Rainfall and Dissolved Oxygen

There were significant correlations between rainfall and DO concentration along the entire Harbour longitudinal axis (**Figure 8**). However, the correlation's strength, significance, and direction varied with depth.





In the surface waters and through the mixing zone, the correlation between rainfall and DO concentration was positive
(increased rainfall was observed with increased DO concentrations) and waned with depth (r = 0.6 6m to 8 m below the
surface at stations KR1, C10, and C08) (**Figure 8**). Towards the Gordon River at station WH2, the strongest correlation (r =
0.677, $p$ = 2 x $10^{-17}$) was observed at approximately 10 m depth with the rainfall metric "20 days rain accumulation". At all
stations, the strength of the correlation between rainfall and DO concentration decreased through the mixing zone, eventually

to non-significance and no relationship. Approximately 30 m below the surface, the correlation (though not statistically
significant) was observed to become inversely related to rainfall at all stations (more rainfall was observed with lower DO
concentrations closer to the seabed).

At stations whose monthly profiles extend into DWR waters (*i.e.* KR1, C10, C08), the relationship between rainfall and DO

concentration strengthened and became statistically significant, though inversely related (high rainfall observed with low DO
concentrations), with depth (**Figure 8**). The strongest correlation along the seabed (r = -0.4238, $p$ = 2.3 x $10^{-5}$) was observed
using 20-day rainfall accumulation at station C10, 3 m from the seabed. At station C08 the strongest correlation (r = -0.3637,
$p$ = 0.00034) in the basin was observed using 30-day rainfall accumulation, 1 m from the seabed.

**3.5 Rainfall and Salinity**

As observed in rainfall-DO relationships, there were significant correlations between rainfall and salinity, whose strength,
significance, and direction varied with depth. There were significant inverse relationships between rainfall and salinity (*i.e.*
more rainfall associated with lower salinity) that reached a maximum strength (r = -0.6) 10 m below the surface for stations
near the ocean endmember. In shallow water, these relationships became stronger (r is observed to approach 0.8 at CH5)

closer to the Gordon River (**Figure 8**). The relationship between rainfall and salinity rapidly weakened in the mixing zone to
statistical insignificance.





Figure 8: Pearson correlation coefficients (r) between rainfall and dissolved oxygen (top) and rainfall and salinity (bottom) through the water column at 1m vertical resolution. Selected rainfall metrics include accumulation 30 days prior to sampling (circles), 20 days (triangles), 15 days (diamonds), 10 days (asterisk), and 5 day (x) accumulation, and 30 day average daily rainfall prior to sampling (squares). Red symbols denote statistical insignificant correlations (p < 0.05).




### 3.5 Southern Annular Mode and Dissolved Oxygen

There were depth specific statistically significant Pearson correlations between monthly SAM index values and DO
concentration at each sampling station (**Figure 9**). In the surface waters the correlations were negative and became
increasingly more positive with depth. The strongest relationships in the surface waters occurred at stations closest to the
mouth of the Harbour (except for station KR1, nearest the King River mouth). The strength of the correlations weakened
with depth and eventually displayed positive r values (indicating the opposite relationship). The DO at stations nearest the
sill responded most strongly to SAM index values along the seabed, with statistically significant positive correlations (high
SAM index associated with higher DO concentration) observed at stations C10, C08, and KR1.

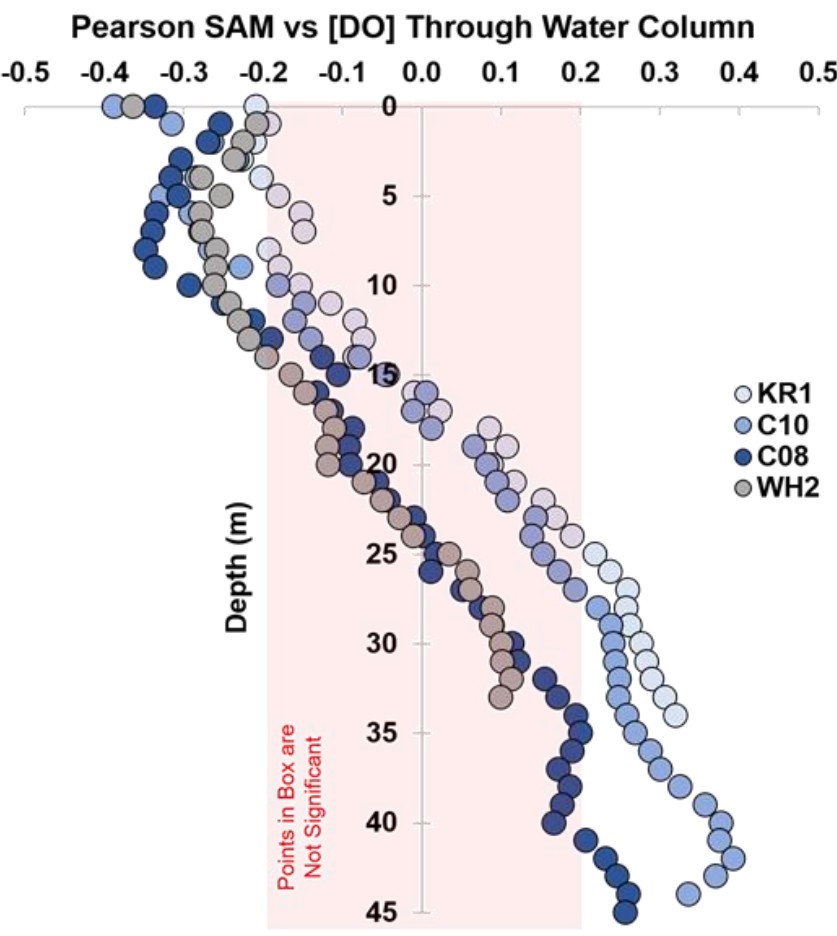

**Figure 9: Pearson correlation coefficients (r) between Southern Annular Mode (SAM) Index and dissolved oxygen concentrations
at KR1, C10, C08, and WH2 through the water column at 1m vertical resolution. Red box denotes statistical insignificant
correlations while all points outside of the box indicate statistically significant correlations (p < 0.05). Note points reference water
depth not height off seabed.**



## 4 Discussion

### 4.1 Rainfall, Oxygen Distribution, and Implications for Green House Gas Production

The water column structure in Macquarie Harbour is similar to that of many fjord and fjord-like DCIs distributed throughout the world's mid to high latitude coastlines. The sill at the system's mouth sets up a strongly stratified water column isolating

basin water from the surface (Inall and Gillibrand 2011; Calvete and Sobarzo 2011). The largest freshwater source, the Gordon River, discharges dissolved organic carbon loads (often over 500 tonnes per day; **Figure 4; also see Appendix Figures A1** and **A2** for context) limiting light penetration to within the first few meters of the water column. These loads restrict photosynthesis in the system to the surface layer (demonstrated by rapidly attenuated chlorophyll-*a* with surface concentrations of 4 to 5 µg L$^{-1}$ and generally undetectable levels at 12 m; see **Appendix Figure A3**).


OC and ON loading exhibited seasonality with significantly greater loadings in winter compared to summer (**Figure 4**). These loads are products of both greater flow volume and significant increases in the measured concentration of OC and ON (**Figure 3**).

The lack of light penetration coupled with limited basin water exchange and large riverine OM loads makes the subsurface waters of this system, and many other similar fjord-like DCIs, naturally prone to oxygen-poor conditions (see Gonsior *et al.* 2008; Bianchi *et al.* 2020). The distribution of dissolved oxygen in Macquarie Harbour is closely tied to the physical separation of the water masses. DO is greatest in surface waters near system endmembers (**Figure 5**), while hypoxia forms regularly and with varying intensity for extended periods in its basins. Below the freshwater lens, DO < 4 mg L$^{-1}$ occurs in

>90% of observations and hypoxia (< 2 mg L$^{-1}$) was observed more than 80% of the time (stations C08 towards WH2; **Figure 6**). Suboxic (< 1 mg L$^{-1}$) concentrations are observed at every station, but more often (up to 36% of the time) at stations closest to the Gordon River. Suboxic conditions have persisted for up to 2 years (**Figure 7**).

The primary mechanism by which basin water is re-oxygenated is the intrusion of ocean water over the sill (*i.e.* DWR).

DWR occurs regularly (see Hartstein *et al.* 2019) but according to our analysis only 5 to 7 times per decade for volumes significant enough to relieve suboxic conditions up-harbour (see **Figure 7**). The drivers of DWR were examined in Hartstein *et al.* (2019) and included sustained N and NW winds and low (< 990 hPa) atmospheric pressure. One of the DWR drivers not entirely resolved was the role of freshwater inputs into the system. Our analysis of rainfall and DO distribution suggests that during periods of high rain (*i.e.* winter), DO tends to be lower just above the seabed (**Figure 8**). During periods of low

rainfall (*i.e.* summer), DO above the seabed is higher. Correlations between rainfall and DO concentration were not statistically significant through a large portion of the sub-halocline, but with depth became statistically significant a few meters above the seabed (*see* **Figure 8**). This depth-dependent relationship between rainfall and DO suggests that the amount of river water entering the system is an important driver of DWR and thus the distribution of DO.



Rainfall in the Macquarie Harbour catchment is influenced by the oscillations of the Southern Annular Mode (SAM). Our analysis also showed that SAM index values significantly correlated with DO conditions in the surface and near-bottom layers. Positive SAM index values were associated with lower DO in the surface waters and higher oxygen concentrations in the basins nearest the sill (**Figure 9**). Positive SAM index values indicate high pressure in the southern hemisphere mid-latitudes, leading to weaker westerly winds (Taschetto and England 2009; Dey *et al.* 2018). For the West Coast of Tasmania,

weaker westerly winds result in less moisture captured by the mountain ranges that form the eastern edge of the Harbour's catchment and subsequently, less rainfall occurs. Less rainfall in the catchment means less river flow to the Harbour, which promotes DWR and relieves deep basin hypoxia. Austin and Inall (2002) observed similar broad-scale climate oscillations (in this case, the North Atlantic Oscillation) affecting the basin water in Scottish Sea Lochs.

Rainfall metrics and the effect of SAM only significantly correlated with surface waters and the deeper basin waters of the Harbour (**Figures 8** and **9**). Without a direct supply of oxygen from either physical mixing or photosynthesis, the slow but steady uptake of DO by microbes (into heterotrophic (*e.g.* aerobic respiration) and autotrophic (*e.g.* ammonia oxidation) pathways establishes oxygen-poor conditions (Maxey *et al.* 2020). The low light and suboxic conditions often present in the sub-halocline up Harbour likely promote DO uptake by nitrifying microbes. Depending on the amount of available DO, the

eventual end product of ammonia oxidation (first step of nitrification) may differ (Arrigo 2005; Codispoti *et al.* 2005; Lam *et al.* 2011). At suboxic concentrations, ammonia oxidation begins to favour the production of $N_2O$ (a potent greenhouse gas; GHG) over $NO_2^-$ (Goreau *et al.* 1980; Ji *et al.* 2018; Frey *et al.* 2020;). Da Silva *et al.* (2021) observed significant differences in the microbial communities present in the Harbour's basin waters, with communities comprising a more significant proportion of GHG producers than in the surface waters. Our improved understanding of the drivers and oxygen distribution

in the system seems to corroborate their findings. Their observations of community composition and our DO distribution observations in the Harbour suggest that the basins' communities are primed for GHG production during periods of prolonged suboxia.

The eventual fate of riverine OC and ON entering the harbours basins will depend upon the oxygen conditions present.

Suboxic DO conditions favour the production of GHG such as $CH_4$ and $N_2O$, and the lack of large DWR events promotes suboxic basin water conditions. High freshwater inputs are associated with low basin water DO, and we postulate that this is due to the impediment of DWR. In Macquarie Harbour and DCIs with constricted inlets and large rivers, the physical effects of rainfall or river flow (as driven by broader climate forcings like SAM) may be tied to the eventual fate of C and N entering the system (**Figure 10**).





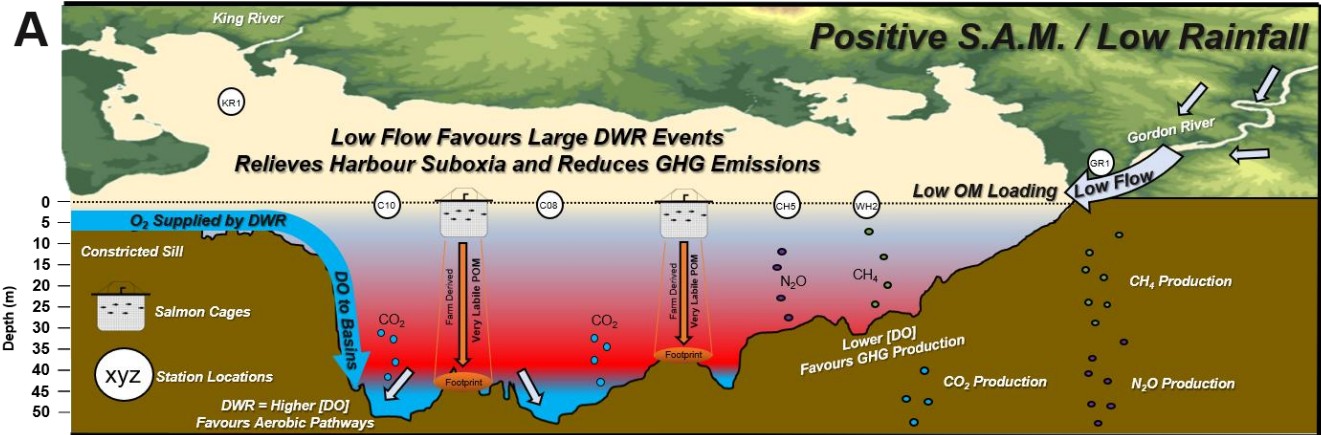

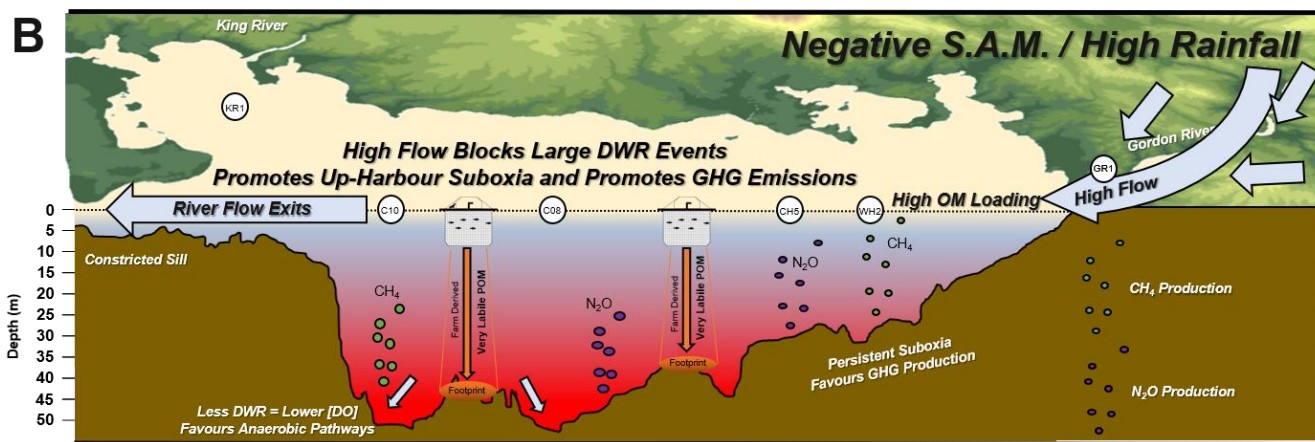


**Figure 10: Conceptualisation of the role of freshwater inputs on the distribution of DO and potential for GHG production in the Harbour. (A) Harbour DO and GHG during periods of low rainfall (or positive SAM index) promoting DWR. (B) High rainfall (negative SAM index) blocks DWR and promotes the spread of suboxia and thus GHG production.**

### 4.2 Climate Change Predictions for Tasmania's West Coast and Dissolved Oxygen

Climate change predictions for the west coast of Tasmania suggest that there will be greater and more intense rainfall in the winter and lower rainfall in summer (Viney *et al.* 2009; Grose *et al.* 2010; Bennett *et al.* 2010). Increased rainfall means greater river flow and OM loading (**Figures 2** and **3**) and is associated with decreased basin water DO in this system (**Figure 8**). High winter river flow may reduce the chances of significant DWR events and promote prolonged basin water suboxia (**Figure 10**).



While large DWR events have occurred during both summer and winter, they are infrequent (*see* Hartstein *et al.* 2019). Increased winter river flow may reduce the chance of DWR occurrence by blocking marine intrusions over the sill; this would result in more extended periods of basin water suboxia and GHG production. An analysis of the weather station wind
data available from 1993 to 2014 at Cape Sorell (near Hell's Gates Inlet; see **Figure 1** for location) shows that wind conditions required for DWR (sustained N and NW winds; Hartstein *et al.* 2019) are more frequent (over 30% of the time) in winter compared to summer (18% of occurrences). If the climate predictions discussed in Grose *et al.* (2010) and Bennett *et al.* (2010) are realised, then this may result in a disproportionate reduction in significant DWR events for the Harbour and increased GHG emissions. Likewise, if winter rainfall is reduced, DWR may become disproportionately more frequent, thus
reducing potential GHG emissions.

It is the orientation of Macquarie Harbour that makes N and NW winds a significant driver of DWR. Sustained winds from this direction push water up the Harbour's longitudinal axis, creating wind set up at the head of the system, and obliges marine water to cross the sill (Hartstein *et al.* 2019). Upon entering the main harbour body, the oxygen rich dense marine
water mass sinks into the deepest basins. If climate change affects the frequency and duration of wind occurring in N and NW directions, DWR (and GHG emissions) may be affected through an additional mechanism independent of rainfall (*e.g.* changes in wind direction).

There is some evidence that climate forcings (like SAM) have been significantly strengthening winds and waves along
Tasmania's west coast since the 1970s (*see* Hemer 2010a and 2010b; Kirkpatrick *et al.* 2017; Marshall *et al.* 2018; Sharples *et al.* 2020). If climate driven increases in wind speed result in changes to the N and NW wind patterns, then this may have implications for DWR frequency in the Harbour. Likewise, increased wind and wave forcings have been suggested as a cause for further sedimentation of the harbour mouth (Sharples *et al.* 2020), which would constrict the sill and impede DWR. At the moment, the confluence of large-scale climate drivers (*e.g.* increasing SAM; climate driven winds and rainfall
patterns) make predicting future DWR patterns (and the resulting GHG release) in Macquarie Harbour a difficult task.

In other fjord and fjord-like DCIs, the predicted impact of climate change on rainfall varies, as does each system's sensitivity to freshwater input for DWR. In scenarios where freshwater input is the important driver of bottom-water oxygenation (for instance, Lochs Etive (in Edwards & Edelsten 1977; Austin and Inall 2002; Gillibrand *et al.* 1995) and Ailort (in Gillibrand
*et al.* 1996) increased freshwater supply may increase the extent and duration of hypoxia and thus GHG emissions.

One of the critical factors buffering GHG releases from these systems is the well-oxygenated surface water layer. Some portion of the dissolved $CH_4$ produced in sediments and suboxic regions of the Harbour may be oxidised upon reaching more normoxic regions (Reeburgh 2007). In many systems the relatively high concentrations of $CH_4$ produced in deeper anoxic
portions of the system are observed to be reduced in surface layers (*e.g.* Storfjorden in Mau *et al.* 2013; Pearl River Estuary





in Ye *et al.* 2019; Saguenay Fjord in Li *et al.* 2021). Nevertheless, the concentration of $CH_4$ at the surface can often be observed at supersaturated concentrations and thus will still be sources of $CH_4$ to the atmosphere. The ebullition of GHG produced in sediments can also shuttle $CH_4$ to the surface before it is fully oxidised. This process is heavily influenced by physical factors such as sediment grain size (higher magnitudes in finer sediments) and hydrostatic pressure (Liu *et al.* 2016;

De Mello NAST *et al.* 2018). In Macquarie Harbour, and many DCIs, the areas most prone to suboxia are areas where the underlying seabed is composed of fine-grained organic-rich riverine sediment (Carpenter *et al.* 1991; Teasdale *et al.* 2003). Additionally, the water level (and thus hydrostatic head) in the Harbour and many other systems varies with freshwater supply and would also contribute to the ebullition of $CH_4$.

Whether the magnitude of climate-induced GHG emissions from DCIs are significant enough to accelerate further climate change (and thus further accelerate GHG emissions through a positive feedback loop) (**Figure 10**) remains to be seen. While the oceanographic mechanisms by which fjord-like DCIs operate are similar, their unique morphologies and surrounding landscapes / hydrologies must be better understood to account for their potential as sources of GHG under future climate scenarios. In order to better manage resource use affecting these systems (*e.g.* hydroelectric dams, salmon farms, discharge

outfalls), understanding the connection between the physical processes (such as freshwater supply, weather patterns, DWR) and biogeochemical processes (pelagic oxygen demand, $CH_4$ and $N_2O$ production, $CH_4$ oxidation) needs to be established.

### 4.3 Harbour Management Implications

Resource use in Macquarie Harbour is similar to that of many DCIs. The relatively deep, cool, and oxygen-rich sheltered

waters provide an excellent location for sea cage salmon aquaculture (Gillibrand and Turrell 1997; Skogen *et al.* 2009; Inall and Gillibrand 2010). Sea cage culture requires the fish to be immersed in their environment, and in Macquarie Harbour the areas where farms are situated span some of the area over the deepest basins (**Figure 1**). Significant DWR events, like those occurring in May 2015, can cause physical disturbances to water column structure (*e.g.* internal waves) that significantly affect fish farms. The May 2015 event, for example, caused internal waves that transported oxygen-poor bottom water to the

surface (Hartstein *et al.* 2019). Because rainfall affects DWR, the predicted increase in winter intensity may reduce the chances of significant DWR events and thus reduce the likelihood of adverse physical disturbances to fish farms. On the negative side, farm-derived feed and faeces waste represent a highly labile source of OM to the seabed.

Farm waste footprints may be hotspots of GHG production, but the magnitude of that production is still largely undescribed.

If climate change reduces the frequency of DWR, the processing of farm-derived OM may be less efficient as less oxygen will be available to fuel aerobic respiration (Brooks *et al.* 2000; Pereira *et al.* 2004). This may lead to possible increased GHG production and will facilitate the spread of *Beggiatoa* matting (Crawford *et al.* 2001; Crawford 2003). Presently, no





studies describe the rate of the GHG output under salmon farms in any of these systems, leaving open a significant knowledge gap for future aquaculture and GHG research.


Rainfall is not the only driver of river flow into the Harbour. The Macquarie Harbour catchment has two upstream hydroelectric dams regulating flow into the main rivers. Dam release water introduces additional POC and DOC load, especially during low reservoir periods (MHDOWG 2014). These releases may impact DWR by blocking events that might otherwise have occurred. Given the infrequency of large DWR events and their important role in mitigating basin water

suboxia, it seems prudent for hydroelectric management to consider the implications of ill-timed releases on harbour health and their subsequent impacts on GHG production.

## 5 Conclusions

In summary, rainfall significantly affects OM and nutrient concentrations entering Macquarie Harbour. Importantly, rainfall

is seasonal and has a significant depth specific impact on the distribution of dissolved oxygen in the harbour body. It appears that the impact of rainfall on basin water hypoxia is driven by physical forcings, namely the impediment of DWR.

The Southern Annular Mode (SAM) climate oscillation index also significantly correlates with DO concentrations in the surface layer and a few meters above the seabed. However, the direction of the correlation is layer-specific. The SAM index

is highly variable but has been increasing in recent decades. Climate change is predicted to result in wetter winter / drier summers for the Tasmanian West Coast, resulting in fewer DWR events in winter and more frequent and intense DWR events in summer.

Currently, there is no information describing the distribution of $CH_4$ and $N_2O$ in Macquarie Harbour or how this varies with

river loading. However, the DO distribution suggests that most of the production will be near the Gordon River mouth (*e.g.* stations CH5 and WH2). If DWR is stymied in winter by increased rainfall, the suboxic conditions promoting GHG production may become even more prolonged, especially up-harbour.





## 525   6 Appendices

**Figure for Appendix: (A1) Total Suspended Solids concentrations observed at station GR1. Samples collected 2m from surface (red) and 2m from seabed (black). (A2) Total organic carbon (solid lines) and dissolved organic carbon (dashed lines) concentrations observed at station GR1. Samples collected 2m from surface (red) and 2m from seabed (black). (A3) Chlorophyll-a**
**collected from stations KR1 (black), C08 (green), and WH2 (blue). Samples collected 2m from surface (solid lines open symbols) and 12m from surface (dashed lines closed symbols). All data displayed reflect dates from September 2014 to November 2021 and are used here as an example of Macquarie Harbour water column properties.**




## 7 Data Availability

This data set is not available for the public.

## 8 Author Contributions

**Johnathan Daniel Maxey** – *Conceptualization, Field Collection, Analytical Methodology, Data Analysis, Writing – Original Draft, Writing – Review & Editing*


**Neil David Hartstein** – *Conceptualization, Field Collection, Analytical Guidance, Writing – Review & Editing, Funding*


**Aazani Mujahid** – *Writing – Review & Editing*

**Moritz Müller** – *Analytical Guidance, Writing – Review & Editing, Funding*

## 9 Acknowledgements

We would like to thank the Analytical Services Tasmania for their help in clarifying methods descriptions and general discussions about the implications of analytical methods. Appreciation to Hydro Tasmania and the Bureau of Meteorology for providing links to rainfall and river flow data within the Gordon River catchment. We want to thank the vessel operators for their tireless efforts during the many inclimate weather days spent with us on the Harbour (Sam and Sean Gerrity, the late Trevor Dennis, Torsten Schwoch, and Ryan Gunton) and the many technicians responsible for helping to collect portions of
this field dataset (Justyn, Dora, Grace, Shukry, Amirul). We would also like to thank our families for supporting those long days away from home and hours spent in front of the screen to get this information and its implications published - we had to borrow the time from you, and we are grateful. This research has been supported by the Newton-Ungku Omar Fund (grant no. NE/P020283/1).

## 10 Competing Interests

The authors declare that they have no conflict of interest.



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
