# Peer review of "The influence of mesoscale climate drivers on hypoxia in a fjord-like deep coastal inlet and its potential implications regarding climate change: examining a decade of water quality data"

_Biogeosciences, 2022_

## Author Response (AR1)

**Reviewer 1**

**General Comments:**

**These comments were taken directly from the discussion. After receiving the comments we have revised the manuscript accordingly. Our original replies to the reviewer comments are shown in red text. Our post-revision comments are shown in blue text.**

*We thank the reviewer for their positive feedback of this draft manuscript. The comments were insightful and provide an avenue for further clarification and improvement to the final version of this article.*

*We have highlighted several of your comments below for further discussion, but note we will address each of the comments in full in the revised manuscript along with feedback from any additional reviewers.*

*Below we briefly address your comments (bold italicised font) in hopes that it might stimulate deeper discussion within this forum.*

I found this to be a well-structured analysis of a complicated system with clear results to support the authors' hypotheses. The authors describe the system and all of the potential drivers and contributors of observed low oxygen in detail and leverage a unique, long-term dataset to do so. The figures were of high quality and supported the statements made by the authors well and it was clear that they are well-acquainted with the relevant literature for this system. I found their conclusions relating changes in rainfall to deep hypoxia and deep water renewal event frequency to be very convincing, though I do wonder what role eutrophication from the increased DOM resulting from large rain events might have – potential positive feedbacks?

*Our analysis shows that with increased rainfall, DOC and DON loading increases as a result of both increased river flow as well as increased concentrations of DOC and DON present in the water.*

*The majority of the additional OM entering the system is primarily dissolved and likely retained in the surface lens where dissolved oxygen concentrations are at their highest. It is important to note that Macquarie Harbour is a "black water system" with relatively low chlorophyll-a concentrations at the surface and undetectable levels below the halocline (in this case 12m depth). The contribution of phytoplankton production to eutrophication would be limited by poor light availability.*

*Undoubtably some river-derived OM will reach the dark sub-halocline layers and depending on its lability will contribute to the removal of dissolved oxygen from the water column through respiratory processes. However, in a previous study (Maxey et al. 2020) the effect of increased riverine OM loading on the rate of water column oxygen demand was not significant.*

*In regard to establishing positive feedback loops (we assume you mean anoxia promoting the generation of CH4 and N2O in the system and further exacerbating climate driven river loading) we feel that this remains a key knowledge gap in the literature (especially for southern hemisphere systems) and that more fjords and fjord-like systems should be investigated to better quantify the significance of this on global scales.*

*In the text we have clarified and alluded to the additional eutrophication from the river (along with its potential effects on hypoxia). We begin to address this through an additional a statement located on lines 136 to 139. In the discussion section we have added a new paragraph dedicated to addressing this comment from the reviewer (lines 427 to 439). This paragraph was derived from our response to the reviewer's statement in the open discussion.*

Aside from some more specific comments (see next section), my only note is regarding the connection drawn between the increased hypoxia and outgassing of greenhouse gases. While the authors provide evidence from the literature to support this hypothesis, I think their claims would be better supported with quantitative measurements to show that in this particular system, this outgassing already occurs and might increase.

*These measurements are ongoing as part of a body of PhD research and will be submitted as a separate publication.*

*We received similar comments from the second Reviewer and have decided to remove the emphasis of Greenhouse Gas Emissions from the Title, Abstract, Research Objectives, but have decided to retain some of our hypotheses in the discussion section where these kind of inferences are more appropriate.*

Overall, this paper appears to fill a notable gap in knowledge for this system and sets up the potential for future analyses on additional questions raised.

**Specific Comments:**

*We thank the reviewer for these specific comments and will address this once we receive feedback from the second reviewer. We have highlighted a few comments for further discussion.*

- Citation for the statement on lines 95-96?

*We have added the following citations to the end of the statement:* Where or when the direct effects of these processes wane, diffusive mixing and water column oxygen demand become the key drivers of oxygen availability (Inall and Gillibrand, 2010; Hartstein *et al.,* 2019; Maxey *et al.,* 2020). "

- Figure 1: In inset map of Tasmania, put box around area that is zoomed in on in larger figure? Also in right map, it is hard to tell where the river is – can you draw a line or something to highlight its path rather than the two arrows?

*We have inserted the box as suggested and have highlighted the river path as suggested*

- Line 155**:** I am not clear on how distinct functional groups support that external climatic drivers influence harbour processes.

*The combination of harbour morphology and external climate drivers (i.e. rainfall patterns, sea level, air pressure, wind direction and speed, etc) establish horizontal and vertical salinity, density, light and nutrient availability gradients in the harbour.*

*The results presented in DeSanto et al. (2020) show that microbial community composition shifts along the gradients as they exist in Macquarie Harbour. Thus, it is reasonable to assume that environmental gradients are key drivers of microbially mediated biogeochemical processes (e.g. production of $N_2O$ and $CH_4$).*

*We will clarify this in a revised version of the manuscript.*

*We have clarified the statement, it now reads (from 160):*

"They found that DWR is the major driver of bottom water oxygen distribution and that Gordon River organic loading is the primary driver of pelagic oxygen demand (POD). **These previous studies demonstrate that external physical drivers (*i.e.* rainfall patterns, sea level, air pressure, wind direction and speed, etc) establish horizontal and vertical salinity, density, light and nutrient availability gradients in this system.** Da Silva *et al.,* (2021) examined the microbial communities present in Macquarie Harbour's water column. They showed distinct functional groups along the harbour's salinity and depth gradients, suggesting that physical drivers may also influence harbour microbial processes."

- Line 162: At this point, I was curious to know how many basins there were in the harbor, how deep they were, etc. and was curious if there was a map or drawing of them. I see later in Figure 10 this is shown, but it may be good to have another figure earlier showing this since these deep basins are a large part of your story.

*We will amend Figure 1 to include a harbour cross section highlighting the basin morphology*

*We have amended Figure 1 to include a harbour cross section highlighting basin morphology and have added a column to Table 1 to indicating the depth of the sampling stations.*

- Line 165: Please add the accuracy/precision of your YSI

*We have added the accuracy and precision as requested in the text*

- Table 1: Perhaps add maximum depth of each station?

*We have added the information to Table 1 as requested*

- Figure 2: Why are there not groupings provided above A and B?

*We have amended Figure 2 to better clarify post-hoc statistical groupings for rainfall and flow by adding colour as well as improving the text placement in the images.*

- For the final publication, note that Figures 5 and 8 are a bit blurry.

*We have attempted to sharpen the images but note that the graphs themselves are comprised of a large amount of data and due to scale issues may appear slightly blurry. We can work with the copy editing team if the improvements are not satisfactory.*

- Figure 7 was really nicely done – good way to display many different variables

*We thank the reviewer for their positive comments*

- Figure 8: Because you have the y-axis crossing at 0 it becomes somewhat hard to tell where one plot ends and the next begins, and also hard to read the axes on plots that cross the y-axis. Perhaps have the y-axis cross at a negative x-value to avoid this and add a dotted line to indicate where 0 is?

*We have amended the figures as suggested. The Y-axis now crosses on the left side of the panel and have added a dashed line at x = 0 to improve figure understanding.*

- Figure 9: same comment about crossing the y-axis as in Figure 8

*We have amended as suggested*

- Figure 10: Really informative figure, curious here about feedbacks of the increased OM loading under high flow – if this will also work to exacerbate low oxygen in combination with the lack of DWR?

*We agree that increased OM loading to the harbour has the potential to exacerbate low dissolved oxygen conditions in the basins, however Maxey et al. (2020) could not resolve the effect of increased DOM loading on oxygen consumption rates in the system's basin waters. There are many reasons why this may be the case, but an obvious reason may be the reduced ability to resolve the effect of OM loading on oxygen demand due to a limited number of measurements (6 months in the case of Maxey et al. 2020). The rates of oxygen demand in the basin waters of this system are relatively low, but despite this hypoxia forms regularly and for prolonged periods due to limited basin flushing.*

*We think this comment is related to the first comment and have added a new paragraph dedicated to addressing this comment from the reviewer (lines 427 to 439) as well as an addition to a statement deeper into the discussion (Line 481):* "High winter river flow may reduce the chances of significant DWR events and promote prolonged basin water suboxia by stimulating pelagic oxygen demand through increased riverine OM loading (Maxey *et al.,* 2020; **Figure 10**). "

- One other thing to consider is that deoxygenation of the deep waters outside the harbor will also decrease the O2 available in the water coming up during these DWR events, so this may also further inhibit relief from low oxygen?

*We are unaware of any evidence that suggests this may be a contribution factor to the deoxygenation of Macquarie Harbour's basins. In fact, in Hartstein et al. (2019) rapid changes in basin water dissolved oxygen (as well as temperature) was used to detect DWR. The West Coast of Tasmania is still relatively pristine, and it would be interesting to understand how the DOC-rich harbour water might be affecting dissolved oxygen concentrations in the coastal ocean as is exits the system. To date, this remains unresolved.*

*We have addressed this comment by adding to the discussion section on Lines 514 – 516:* "Less frequent DWR may be further exacerbated by coastal deoxygenation (see Keeling *et al.,* 2010; Levin and Breitburg, 2015; Wang *et al.,* 2017; Gupta *et al.,* 2021). In such systems with coastal deoxygenation, the oxygen mass introduced to basins will be lower, further promoting hypoxia and potential GHG release." *We thank the reviewer for helping us draw some global relevance for this research.*

- Data Availability: Will the dataset be made available following publication? For transparency and ethical scientific practices, the data used should be made public.

*We plan to make the data available upon the completion of the PhD research.*

**Technical Corrections:**

- Title: "its" should be "the" or "their"

- Line 22 : "predicts"

- Line 62: "it" should be "they" or "these factors"

- Line 90: "it's" should be "the"

- Line 95: Please define DWR before using acronym (was defined on line 45)

*We have addressed these technical corrections as requested.*

**Reviewer 2**

**These comments were taken directly from the discussion. After receiving the comments we have revised the manuscript accordingly. Our original replies to the reviewer comments are shown in red text. Our post-revision comments are shown in blue text.**

The major challenge of the authors' work is to tease apart seasonal and inter-annual climate variations affecting the organic matter (OM) loading and hypoxia formation in a deep coastal inlet. Considerable amount of observational data is acquired and statistically processed to address three issues (in line 76 – 86): (1) effects of rainfall on OM loading and oxygen distribution; (2) effects of climate forcing on rainfall patterns and associated hypoxia formation; (3) implications on greenhouse gas emissions in this seasonally hypoxic system. Overall, I find issue #1 is well demonstrated, #2 is logically sound; and #3 is loosely based on current dataset. Nevertheless, the topic is interesting and, once the manuscript is improved, it will be suitable for publication in Biogeosciences. The following major issues are suggested for the authors to consider in the next round of revision.

(1) I am not sure whether the rainfall pattern shows seasonal variation? I am very confused with the 8 panels in figure 2, because the authors did not describe any panel (A through H) at all. Is it possible to have a simpler version of figure 2, and demonstrate the rainfall pattern?

    a. *Post hoc testing substantiated the seasonal variation though we will improve and further clarify this in the figures, the figure caption, and in the text.*

    *Specifically, we will improve our in-text references to specific figure panels and will consider making use of colour to better highlight post-hoc groupings.*

We have revised the manuscript to specifically reference each of Figure 2's panels in the text (see lines 232 to 253). We understand that the *Post Hoc* statistical groupings were not satisfactorily highlighted and have revised them to include coloured panelling (to highlight statistically significant seasonal relationships between rainfall and estimated flow) as well as inserted text descriptions of those relationships above each panel.

(2) In figure 3, what is the meaning of x-axis? Does higher values represent more rainfall? My intuition is that, more rainfall results in higher river flow; but why would the Pearson corr. different towards the left of the two panels (at low rainfall and low river flow)?

    a. *The x-axis refers to the number of days used to determine rainfall / flow volume when examining the relationship between this volume and the concentration of DOC/DON.*

    *Higher x-axis values represent a longer period of consideration when calculating the accumulated volume of water either falling into the catchment or estimated to be flowing through the Gordon River.*

    *The Pearson correlation will differ due to improved signal to noise ratios when considering rainfall volume / estimated flow over periods longer than a couple of days.*

    *The strongest correlations between rainfall and DOC/DON concentration were found when considering rainfall over the 5 day-period just prior to sampling for DOC/DON.*

    *The strongest correlations between estimated flow and DOC/DON concentration were found when considering the total accumulated flow 2 to 3 days prior to sampling DOC/DON concentrations.*

    *Our flow estimates are based on rainfall volume and we believe the improvement in the Pearson Correlation when considering accumulated volume over the few days prior to sampling is likely related to hydrological phenomena (e.g. the time it takes catchment runoff to reach the mouth of the Gordon River).*

We understand that some clarification was needed for Figure 3. To address this, we have revised the labels on the X-axes of each panel, as well as added some further description to the figure caption.

(3) In figure 4, the upper panel show no significant seasonal variations in organic carbon loading; in figure 10, why OM loading is low during positive SAM? Can the authors show correlation between SAM and OM loading to support this claim? In addition, the daily average farm carbon load is much lower than riverine input; I would suggest the upstream dams are a much more important factor to consider because dams may dampen seasonal variabilities of river flow and OM loading.

*a.* **We will improve the panel figures to better align with the statistical relationships (including these significant seasonal effects) mentioned in the text.**

**In figure 10, OM loading is low during positive SAM index periods because rainfall volume is lower during these periods. The data shown in Figure 3 indicates that greater rainfall is associated with higher concentrations of OC and ON at the Gordon River mouth. Higher OC and ON concentrations + greater river flow yields higher river OC and ON loading.**

**Positive SAM index values are associated with less rainfall in Western Tasmania. Less rainfall in the catchment will result in lower riverine OM loading. We will consider adding an additional figure to the manuscript (possibly as a panel) to show the relationship between SAM index and OC and ON load.**

**We agree that upstream dams have the potential to seriously impact the river flow and subsequently the OM load entering the harbour but note that there is a large amount of catchment area (and resulting runoff volume) that remains unregulated by the dam. Figure 2A does show what appears to be this dampening effect (especially when compared to Figures 2C and 2D), and we believe this could be one of the reasons that the correlations presented in Figure 8 are not stronger.**

We understand that the *Post Hoc* statistical groupings were not satisfactorily highlighted in the figure nor in the text, and we have revised them to include coloured panelling to highlight statistically significant seasonal relationships between OC and ON Loading (we have also amended the figure caption).

A very brief amendment to the text was added to Line 304 to better clarify the statistical significance of the seasonality.

We thank the Reviewer for their comment regarding the effect of the upstream dam on OM loading, and have added further description of this process in the Study Site description on Lines 136 -139.

This manuscript does not present any greenhouse gas data; with these data the manuscript would have been more convincing by linking the greenhouse gas formation to SAM and further to climate variation. The aim #3 of this manuscript remains unresolved.

b. ***Our goal was to establish that the conditions for the formation of GHGs exist in the harbour (sub-oxia), establish how these conditions are spatially distributed, and when these conditions are likely to occur. We stated the 3$^{rd}$ aim of the paper as:***

*"Discuss implications for managing these systems regarding the regulation of freshwater input, OM loading, and the potential for GHG emissions".*

***Our intention is to draw the link between the formation of oxygen poor conditions and the resulting potential for these systems to generate greenhouse gasses such as $CH_4$ and $N_2O$. We never intended to show the distribution of GHG in this system in this paper, only that the potential exists and should be studied further.***

***We will leave the final decision of how to best address this concern to the editorial team, but we do take this concern seriously and note that the previous reviewer also shared similar concerns.***

Both Reviewer 1 and Reviewer 2 have made the same comment. We have removed the emphasis to Greenhouse gas emissions from the Title, Abstract, Research Objectives. We have retained some of our hypotheses in the discussion section where it is more appropriate.

---

## Referee Report (RR1)

24 May 2022

Referee Report for: **The influence of mesoscale climate drivers on hypoxia in a fjord-like deep coastal inlet and its potential implications regarding climate change: examining a decade of water quality data**

I believe the authors have appropriately incorporated the suggestions and edits from the two reviewers, which has resulted in an improved manuscript. I appreciate the changes to figures and the removal of the GHG emissions commentary. In my opinion, the MS is ready for publication! Great job!